# High CD169 Monocyte/Lymphocyte Ratio Reflects Immunophenotype Disruption and Oxygen Need in COVID-19 Patients

**DOI:** 10.3390/pathogens10121639

**Published:** 2021-12-18

**Authors:** Antonella Minutolo, Vita Petrone, Marialaura Fanelli, Marco Iannetta, Martina Giudice, Ines Ait Belkacem, Marta Zordan, Pietro Vitale, Guido Rasi, Paola Sinibaldi-Vallebona, Loredana Sarmati, Massimo Andreoni, Fabrice Malergue, Emanuela Balestrieri, Sandro Grelli, Claudia Matteucci

**Affiliations:** 1Department of Experimental Medicine, University of Rome Tor Vergata, 00133 Rome, Italy; antonella.minutolo@uniroma2.it (A.M.); vita.petrone01@gmail.com (V.P.); fanellimarialaura@gmail.com (M.F.); giudicemartina94@gmail.com (M.G.); guidorasi@hotmail.com (G.R.); sinibaldi-vallebona@med.uniroma2.it (P.S.-V.); balestrieri@med.uniroma2.it (E.B.); grelli@med.uniroma2.it (S.G.); 2Department of Systems Medicine, University of Rome Tor Vergata, 00133 Rome, Italy; marco.iannetta@uniroma2.it (M.I.); marta.zordan91@gmail.com (M.Z.); sarmati@med.uniroma2.it (L.S.); andreoni@uniroma2.it (M.A.); 3Infectious Diseases Clinic, Policlinic of Tor Vergata, 00133 Rome, Italy; pietro_vitale@outlook.it; 4CNRS, INSERM, CIML, Centre d’Immunologie de Marseille-Luminy, Aix Marseille Université, 13009 Marseille, France; iaitbelkacem@beckman.com; 5Department of Research and Development, Beckman Coulter Life Sciences-Immunotech, 13009 Marseille, France; fmalergue@beckman.com; 6Institute of Translational Pharmacology, National Research Council, 00133 Rome, Italy; 7Virology Unit, Policlinic of Tor Vergata, 00133 Rome, Italy

**Keywords:** cytokine storm, COVID-19, CD169, inflammation, respiratory outcome, T-cell exhaustion, COVID-19 therapy

## Abstract

Background: Sialoadhesin (CD169) has been found to be overexpressed in the blood of COVID-19 patients and identified as a biomarker in early disease. We analyzed CD169 in the blood cells of COVID-19 patients to assess its role as a predictive marker of disease progression and clinical outcomes. Methods: The ratio of the median fluorescence intensity of CD169 between monocytes and lymphocytes (CD169 RMFI) was analyzed by flow cytometry in blood samples of COVID-19 patients (COV) and healthy donors (HDs) and correlated with immunophenotyping, inflammatory markers, cytokine mRNA expression, pulmonary involvement, and disease progression. Results: CD169 RMFI was high in COV but not in HDs, and it correlated with CD8 T-cell senescence and exhaustion markers, as well as with B-cell maturation and differentiation in COV. CD169 RMFI correlated with blood cytokine mRNA levels, inflammatory markers, and pneumonia severity in patients who were untreated at sampling, and was associated with the respiratory outcome throughout hospitalization. Finally, we also report the first evidence of the specific ability of the spike protein of SARS-CoV-2 to trigger CD169 RMFI in a dose-dependent manner in parallel with IL-6 and IL-10 gene transcription in HD PBMCs stimulated in vitro. Conclusion: CD169 is induced by the spike protein and should be considered as an early biomarker for evaluating immune dysfunction and respiratory outcomes in COVID-19 patients.

## 1. Introduction

Coronavirus disease-19 (COVID-19), caused by severe acute respiratory coronavirus-2 (SARS-CoV-2), has led to a global pandemic characterized by high morbidity and mortality. As a consequence of derailed cellular and humoral immune-response activation, numerous individuals develop persistent inflammation associated with a cytokine storm and diffuse organ involvement, mainly associated with severe disease, including acute respiratory distress syndrome (ARDS) [1]. Currently, there are generally accepted clinical guidelines for monitoring infected patients, criteria for hospitalization, several available treatments, and critical-care protocols that have proven effective in reducing mortality [2,3,4], but the heterogeneity of the disease amplifies the need to identify early biomarkers to predict disease progression and guide personalized interventions.

In response to SARS-CoV-2 infection, host cells immediately produce cytokines, including type I interferon (IFN-I), which, in addition to showing antiviral activity, induces the expression of genes involved in limiting the viral spread. After antiviral cytokines are released, sialoadhesin (CD169, also known as SIGLEC-1) is induced and expressed on the surfaces of myeloid lineage cells, such as dendritic cells and monocytes [5,6]. In particular, the two-fold upregulation of CD169 in monocytes (mCD169) exposed to IFNα in vitro has been observed [7].

Previous studies have demonstrated an important role of CD169/SIGLEC-1 in different viral infections, including those from Ebola virus and human immunodeficiency virus (HIV) [8,9,10,11]. Recently, it was suggested that SARS-CoV-2 infects macrophages, particularly CD169-positive macrophages residing in the spleen and lymph nodes, and that this peculiar macrophage cell subpopulation plays a central role in mediating SARS-CoV-2 translocation [12]. Moreover, increased expression of CD169 has been observed in monocytes from COVID-19 patients. In observational studies conducted during the COVID-19 outbreak in France, mCD169 expression, as well as the median fluorescence intensity of CD169 between monocytes and lymphocytes (CD169 RMFI), were also associated with the SARS-CoV-2 infection in patients at hospital admission, underlining its importance as an early infection biomarker [13,14,15,16].

To further investigate CD169 as a contributing factor in SARS-CoV-2 infection and COVID-19 disease, CD169 RMFI was evaluated in COVID-19 patients hospitalized in the Policlinic of Tor Vergata in Rome and correlated with their inflammatory and im-munological statuses, as well as respiratory outcomes.

## 2. Results

### 2.1. Flow Cytometry Analysis of CD169 Expression in COVID-19 Patients and Healthy Donors

Sixty-eight (68) hospitalized patients who tested positive for SARS-CoV-2 RNA (COV) were screened for CD169 expression using flow cytometry and compared to 57 healthy donors (HDs). The MFI of CD169 was not significantly changed in lymphocytes, whereas it increased in monocytes from COVID patients compared with healthy donors (*p* = 0.010) (Appendix A and Table 1). The ratio of the MFI of CD169 between monocytes and lymphocytes (CD169 RMFI) was calculated as described in the Methods section and also referred to in previously published studies [13,15]. Figure 1A shows the distribution of CD169 RMFI in the HD and COV groups, demonstrating CD169 RMFI is higher in patients with COVID compared to HDs, and this difference is statistically significant (Table 1, *p* < 0.001). The accuracy with which CD169 RMFI and CD169 MFI in monocytes (mCD169) could distinguish the COVID-19 patients from HDs was studied via the receiver operating characteristic curve (ROC curve). With a cut-off of 3.01, CD169 RMFI showed an AUC = 0.925 (*p* < 0.001), a sensitivity of 97%, and a specificity of 92%. With a cut-off of 6163, mCD169 MFI showed an AUC= 1, with a *p* = 0.023 (Appendix A, Appendix A), a sensitivity of 100%, but with a lower specificity (83%) when compared to CD169 RMFI).

As shown in Figure 1B, 94.73% of the HD samples were confirmed negative. Among COV, 79% of the patients (*n* = 68) were positive for CD169 RMFI, and 20.58% (*n* = 14) were found to have CD169 RMFIs below the cut-off. Pearson’s correlation coefficient analysis revealed an inverse correlation between the days of hospitalization and CD169 RMFI value, confirming the relevance of RMFI CD169 measurement in the early phases of the infection (more than 10 days, compared to 4 days in the positive group; *p* < 0.001, Figure 1C). Of the 14 false-negative samples, 10 were from patients who had been hospitalized for more than 5 days before testing. The remaining four patients presented symptoms consistent with COVID-19 at the time of nasopharyngeal swab testing, but were not confirmed as SARS-CoV-2 positive after the diagnosis of SARS-CoV-2 infection by real-time PCR. The 14 false-negative samples were excluded from the study. Moreover, the CD169 RMFI in COV was inversely correlated with the mRNA levels for the Nucleoprotein (N) and RNA-dependent-RNA polymerase (RdRp) SARS-CoV-2 genes (Figure 1D).

### 2.2. Demographics and Clinical Classification of COVID-19 Patients

We analyzed the clinical statuses at enrollment and the respiratory outcomes of 54 COVID-19 patients who were hospitalized at the Policlinic of Tor Vergata between May and October 2020 (Table 2 (a) and (b)).

The cohort of patients was divided into two groups, paucisymptomatic (PS) and symptomatic, according to their clinical features on hospital admission or study enrollment for those already hospitalized because of pre-existing diseases. Among the patients, 19 were symptomatic with few clinical manifestations, including at least one COVID-19-related symptom, such as a cough or fever, but not showing signs of pneumonia on physical examination, and therefore defined as paucisymptomatic (PS). Four PS patients showed no symptoms related to COVID-19 and were therefore referred for testing because they were already hospitalized for pre-existing medical conditions; in the PS group, a chest CT scan performed during hospital admission revealed minimal pneumonia in six out of 19 individuals and bilateral interstitial pneumonia (BiP) in six individuals. None of them needed oxygen support at any time during hospitalization.

Of the enrolled subjects in the symptomatic group (*n* = 35), 13 were considered mild (Mild), with typical symptoms of COVID-19 and clinical evidence of pneumonia on physical examination, without showing shortness of breath or dyspnea on enrollment; 12 were considered moderate (Mod), with typical symptoms of COVID-19 and clinical evidence of pneumonia on physical examination, shortness of breath or dyspnea on enrollment, and a saturation of oxygen ≥94% in room air; and 10 were considered severe (Sev), with typical symptoms of COVID-19 and clinical evidence of pneumonia on physical examination, shortness of breath or dyspnea on enrollment, and a saturation of oxygen <94% in room air, or a ratio of arterial partial pressure of oxygen to fraction of inspired oxygen (PaO_2_/FiO_2_) <300 mm Hg or respiratory frequency >30 breaths/min.

Overall, among the 54 COVID-19 patients enrolled, 29 showed radiological signs of SARS-CoV-2-related pneumonia, defined as monolateral interstitial pneumonia (MiP) in six cases and bilateral (BiP) in 23 cases.

Concerning comorbidities, 44/54 patients showed the presence of at least one pre-existing chronic disease, with cardiovascular conditions being the most common comorbidity present (19/54).

At the time of sampling for the CD169 analysis, the mean number of days of hospitalization was 4.54 ± 7.54; most patients were hospitalized for 1–10 days (*n* = 46), six patients were hospitalized for 10–20 days, and two were hospitalized for more than 20 days.

Of the enrolled patients, 23 had been treated with antiviral and corticosteroid therapies, and six died (mortality rate: 8.8%).

The patients in the symptomatic group showed a reduction in the percentage of lymphocytes and higher levels of neutrophils at the time of sampling (Table 2 (b)). Considering inflammatory markers and other parameters of coagulation, the levels of D-dimers, C-reactive protein (CRP), fibrinogen, blood urea nitrogen (BUN), aspartate transaminase (AST) or glutamic oxaloacetic transaminase (GOT), alanine transaminase (ALT), lactate dehydrogenase (LDH), and lipase were all outside the reference ranges and statistically significantly different with respect to the PS group.

Finally, all the COVID-19 patients were evaluated for respiratory outcomes according to the maximum respiratory support received during the hospitalization period. Based on this classification, we divided the patients into two groups: the first group comprised 27/54 patients who did not need oxygen therapy (No OX), who were all PS. The other 27/54 patients received oxygen (OX) in different ways: 12/54 were supported by a nasal cannula (NC) or venturi masks (VMKs), and 15/54 were supported by non-invasive ventilation (NIV), continuous positive airway pressure (C-PAP), and orotracheal intubation (OTI) for invasive mechanical ventilation.

Healthy donors were matched with COVID-19 patients for age and sex: the cohort included 54 COVID-19 patients (median age: 60+/−15; 41 males and 13 females) and 57 healthy donors (median age: 59+/−12; 40 males and 17 females); all the clinical parameters analyzed in the HDs were within the normal reference ranges.

### 2.3. CD169 RMFI Correlates with Biochemical Parameters of Disease Severity and Is Associated with Pneumonia Statuses of COVID-19 Patients

We evaluated the association of CD169 RMFI with laboratory markers and clinical severity in true-positive COV within 5 days of hospitalization (*n* = 54). The CD169 RMFI in COV patients was positively correlated with some biochemical parameters associated with COVID-19 severity, such as fibrinogen, lipase, and GOT (Figure 2A). The analysis of the mCD169 MFI revealed no significant correlation with the same parameters (Appendix A). No other correlations were observed between CD169 expression and inflammatory biomarkers at sampling.

To analyze the association between CD169 RMFI and pneumonia status, the COV group was divided according to chest computed tomography (CT) images at the time of hospitalization and sampling: no pneumonia and non-interstitial pneumonia (None+P, *n* = 28), monolateral or minimal interstitial pneumonia (MiP, *n* = 6), and bilateral or severe pneumonia (BiP, *n* = 23). As shown in Figure 2B, a significantly higher CD169 RMFI was observed in the group with bilateral interstitial pneumonia when compared to monolateral pneumonia (*p* = 0.018) or compared to the None+P group (*p* = 0.031). Although an increased expression of CD169 was observed in the most severe forms of pneumonia, mCD169 MFI was not significant in discriminating the severity of the lung condition (*p* = ns, Appendix A).

### 2.4. CD169 RMFI Correlates with the Expression of Pro-Inflammatory Cytokines in COVID-19 Patients and Is Altered by Therapy

To evaluate the effect of therapy on CD169 expression, patients were stratified based on the drug treatment received (antiviral and corticosteroids, herein referred to as treated or not (untreated)). The group of patients under antiviral and corticosteroid therapy at the time of sampling (treated COV) showed significantly lower expression of CD169 than untreated COV (*p* = 0.033) (Figure 3A, Table 3). No significant differences based on drug treatment were observed when analyzing the CD169 MFI on monocytes (Appendix A). The expression of a selected group of cytokines was also analyzed in the blood samples of COV patients and HDs by qRT-PCR (Figure 3B, Table 3). IL-6, IL-10, IFN-γ, and TNF-α expressions were significantly higher in untreated COV patients than in HDs, while in treated patients, IL-6 and IL-10 had significantly lower values than those of untreated patients (*p* = 0.012 and *p* = 0.001). Moreover, Pearson’s analysis revealed a positive correlation of CD169 RMFI with IL-6 (Rho = 0.415, *p* = 0.015) and with IL-10 (Rho = 0.488, *p* < 0.001) in untreated COV, while an inverse correlation with IL-6 was observed in treated COV (Rho = 0.506, *p* < 0.001). No significant correlation of CD169 RMFI with TNF-α or IFN-γ expression levels was observed (Figure 3C).

### 2.5. CD169 RMFI Correlated with the Expression of T-lymphocyte-Differentiation and Senescence/Exhaustion Markers in Untreated and Treated COVID-19 Patients

The analysis of the T-lymphocyte cell phenotype demonstrated a significant difference in important markers of T-cell differentiation and exhaustion in COV-treated or untreated patients compared to HDs (Table 4).

Among CD4+ T cells, a decrease in the effector memory (EM) subset and an increase in the terminal effector memory (TEM) subset were observed in both treated and untreated COV compared to HDs, and a significant increase in CD4+ cells expressing markers of exhaustion (PD1+) was detected only in treated patients. Among CD8+ T cells from COVID-19 patients, a decrease in the central memory (CM) subset was significant in the untreated group in comparison to HDs, while in the treated group, the observed decrease was not statistically significant. The increase in EM and TEM subsets was present in both patient groups. The percentage of CD57+-positive cells increased significantly in COV patients compared to HDs, as well as in treated compared to untreated COV
.
Furthermore, there is a significant decrease in CD8 NAIVE cells in both untreated and treated patients.

The expression of CD169 was closely related to immunological modifications, especially in untreated patients. Pearson’s analysis revealed a significant correlation between specific markers of CD8+ T-cell differentiation and exhaustion and CD169 RMFI, as illustrated in Figure 4 and Table 5.

Among CD3+CD8+ T cells from untreated COV, CD169 RMFI was associated with a decrease in NAIVE cells (*p* = 0.038) and positively correlated with EM (*p* = 0.016)**.** In treated COV, CD3+CD8+ CM cells negatively correlated with CD169 RMFI (*p* = 0.050), while a positive correlation with TEM cells was observed (*p* = 0.012). Finally, the analysis also revealed that CD169 RMFI positively correlated with the CD57+ senescence marker in CD3+CD8+ T cells in untreated COV. No significant correlation was found between mCD169 MFI and the expression of T-cell markers analyzed (Appendix A).

### 2.6. CD169 RMFI Correlated with the Expression of Differentiation and Maturation Markers in B Cells from COVID-19 Patients

The analysis of the B-lymphocyte cell phenotype revealed a significant difference in markers of B-cell differentiation and maturation in the COV group relative to the HD group (Table 6). Among COV patients, in CD45+CD19+ B cells, the percentage of positive marginal B cells and NAIVE B cells significantly decreased in parallel with a decrease in the numbers of switched memory cells and non-switched B cells, and a significant increase in plasmablasts.

Pearson’s analysis revealed a significant correlation between markers of differentiation and maturation in B cells and CD169 RMFI, as illustrated in Figure 5 and Table 7. In particular, among CD45+CD19+ B cells (Figure 5A), the expression of CD169 was associated with an increase in marginal B cells (*p* = 0.025) and NAIVE B cells (*p* = 0.032) in untreated COV, while in the treated group, no correlation was observed. Switched B cells and plasmablasts negatively correlated with CD169 RMFI (*p* = 0.025 and 0.010, respectively) in untreated COV as well as in treated COV patients. Moreover, a significant inverse correlation between the specific SARS-CoV-2 IgG analyzed in sera and the RMFI of CD169 was observed in both patient groups (Figure 5B). No significant correlations were found between mCD169 MFI and the expression of B-cell markers analyzed (Appendix A).

### 2.7. CD169 RMFI Reflects the Severity Score and Respiratory Outcome of COVID-19 Patients during Hospitalization in Relation to Treatment at Sampling

We then evaluated the CD169 RMFI in relation to the clinical score and its predictive value for oxygen need during hospitalization. As shown in Figure 6A and Table 8, at all clinical scores, patients showed higher CD169 RMFI than HDs. In untreated patients, CD169 RMFI was found to be significantly higher in symptomatic patients than asymptomatic subjects (*p* < 0.008, Table 8). Notably, treated patients with diverse disease scores showed no significant differences in CD169 RMFI, but they were significantly different from HDs (*p* = 0.042). Regarding pneumonia status, CD169 RMFI was markedly increased in untreated patients with bilateral interstitial pneumonia (BiP) compared to patients without pneumonia or no interstitial pneumonia (*p* = 0.045), and to patients showing monolateral or minimal interstitial pneumonia (MiP) (*p* = 0.035). No significant differences in CD169 RMFI were detected when comparing groups of patients with diverse radiological findings, but they were still significant after comparing BiP with HDs (*p* = 0.015; Figure 6B).

To evaluate the predictive value for oxygen need during hospitalization, COVID-19 patients were divided into two categories representing the respiratory outcomes: no oxygen support needed (None) or oxygen support (OX). As shown in Figure 7A and Table 9, CD169 RMFI was found to be significantly higher in the OX group than the None group (*p* = 0.001) only in patients who were untreated at sampling, while no differences were observed in treated patients. Moreover, statistically significant differences in the percentage of senescent CD8+ cells between treated and untreated patients within the OX and None groups were observed (Figure 7B). In addition, significant increases in the transcriptional levels of IL-6 (*p* = 0.050) and IL-10 (*p* = 0.017) were found across respiratory categories in the untreated group (Figure 7C,D). The accuracy of CD169 RMFI in predicting the respiratory outcome in untreated or treated COVID-19 patients was studied by means of the receiver operating characteristic curve (ROC curve) to identify patients requiring respiratory support (Figure 7E). With a cut-off of 48.31 for the untreated group, the sensitivity and specificity at the optimal operating point were 89% and 80%, respectively, with an area under the ROC curve (AUC) of 0.879 (*p* < 0.001), indicating that it is more specific than the other markers analyzed (Table 10).

### 2.8. SARS-CoV-2 Spike Protein Stimulation Enhanced RMFI CD169 in PBMCs from Healthy Donors

To clarify whether CD169 was directly activated by SARS-CoV-2, PBMCs from seven healthy donors were stimulated in vitro for 24 h with increasing concentrations of the spike protein. The viral protein induced the expression of the activation marker HLA-DR in monocytes and CD169 RMFI in a concentration-dependent manner (Figure 8A). In addition, the expression of IL-6 and IL-10 was significantly higher after spike stimulation (Figure 8B). Pearson’s analysis showed a positive correlation between CD169 RMFI and IL-6 and IL-10 expression, confirming the data observed above in COV untreated patients (Figure 8C).

## 3. Discussion

The role of CD169+ macrophages in immune regulation and human diseases has been widely reported [5]. In viral infections, CD169+ macrophages residing in lymphoid organs are the first cells that bind incoming pathogens and act as guardians to prevent their further spread. Circulating monocytes reflect the systemic immune response to infectious agents. In SARS-CoV-2 infection, myeloid cells have been described as responsible for the pathophysiology of the disease by contributing to local tissue damage and acting as potential producers of cytokines that lead to the hyper-inflammatory state observed in severe COVID-19 [17,18,19,20]. Indeed, it was shown that CD169 expression was strongly increased on circulating monocytes from COVID-19 patients compared to those from healthy donors or patients with bacterial sepsis. Increased CD169 expression was also associated with an increase in other activation markers, including CD64, CD68, and CD38 [12]. Moreover, CD169+ monocytes were present in high numbers in the early stages of the disease and in the group with milder cases, where the monocyte compartment consisted almost exclusively of CD169 clusters [21,22].

In accordance with recent work [13,14,15,16], the analysis that we carried out on whole blood taken from COVID-19 patients at the time of admission showed high levels of CD169 RMFI, which correlated with the SARS-CoV-2 RNA expression detected in swabs.

When compared to HDs, the expression of CD169 in COVID-19 patients mainly depends on the high CD169 expression in monocytes, while in lymphocytes no significant differences were found in COV compared to healthy donors. Since the MFI is a very variable value with a high standard deviation in the cohort, the use of intra-patient ratio obtained a higher statistical significance (*p* < 0.001) and high specificity. Moreover, in the clinical routine, it is easier to measure this fluorescence ratio (independent of cytometer settings) rather than cell numbers or mean fluorescence (requiring precise pipetting of venous blood and calibration beads). By analysis of the CD169 monocytes/lymphocytes RMFI, we found a higher ratio in COVID-19 with respect to HD in correlation with the complexity of the immune system dysfunction, inflammatory markers, and clinical aspects, that cannot be found using the MFI alone. The use of the RMFI value increases the robustness of the analysis, repeatability, reproducibility, and accommodates the potential variability of the laser power and detector sensitivity.

In some cases (10/68 patients), the RMFI of CD169 was comparable to healthy donors (false negatives). Of these patients, 10 were hospitalized for more than 5 days, confirming the decrease of the CD169 RMFI during the time post-infection, suggesting the importance of the early analysis of the CD169 marker. These false-negative samples were excluded from the study.

Interestingly, CD169 RMFI was strongly associated with various clinical and biological parameters and reflected not only the patient’s status at admission, but also the progression of the disease.

Notably, a positive correlation of CD169 RMFI with the levels of several pro-inflammatory mediators predictive of severe COVID-19, such as fibrinogen, lipase, and GOT, and the associated cytokine storm [23,24,25], was found in our patient cohort.

Moreover, we also demonstrated that the modulation of CD169 depended on the drugs used for the management of patients, highlighting the need to account for treatments when using CD169 RMFI to monitor COVID-19 patients. Interestingly, we found that CD169 RMFI was strongly correlated with IL-6 and IL-10 mRNA levels, which are known to be associated with poor outcomes [26,27]. CD169+ macrophages play a central role in mediating SARS-CoV-2 translocation in the spleen and lymph nodes, and thus contribute to viral replication and spread, as well as the resulting cytokine storm [11,12,13,14,15,16,17,18,19,20,21,22,23,24,25,26,27,28].

The imbalance in the levels of immune activation and immune suppression is crucial in the loss of host defense against SARS-CoV-2 infection [29,30,31], and changes in the expression of some immune cells are considered to be predictors of severity/mortality in COVID-19 patients [32]. Accordingly, we analyzed the association between CD169 expression in the early phase of SARS-CoV-2 infection and the T- and B-cell immune responses. CD169 RMFI was closely correlated with changes in differentiation markers in CD8+ T cells from COV patients, particularly in untreated patients, and inversely correlated with the expression of CM and EM cells. In fact, in CD8+ T cells, CD169 RMFI was associated with a decrease in NAIVE cells and an increase in EM cells. Notably, CD169 RMFI correlated positively with the percentages of CD8+ CD57+ senescent lymphocytes. It has been demonstrated that CD169+ macrophages that present in tumor tissue are correlated with CD57+CD8+ T-cell or NK-cell infiltration [33]. The CD57 marker has been used to assess functional immunodeficiency in several diseases, suggesting that positive cells do not proliferate, despite the preserved ability to secrete cytokines after activation [34]. COVID-19 patients have increased numbers of CD8+ T cells expressing CD57, which is considered a key marker of senescence and is associated with both human aging and chronic infections; CD57 expression was also recently described as having an association with other endogenous markers [35,36]. We also found an association of CD169 RMFI with PD1 expression, a molecule crucial for the induction and maintenance of peripheral tolerance and T-cell stability and integrity. Indeed, the PD1/PD-L1 axis mediates potent inhibitory signals to block the proliferation and function of T-effector cells, inhibiting antiviral immunity [37].

CD169 RMFI correlated with changes in CD8 and B-lymphocyte immunophenotypes and with some of the inflammatory markers. The analysis did not confirm correlations with the CD4 subset, highlighting the link with CD8 senescence and exhaustion phenotypes, which are specific immune dysfunctions of COVID-19. CD169 RMFI was inversely correlated with the presence of specific SARS-CoV-2 IgG in serum [12,13]. As a novelty of this study, by characterizing the B-cell compartment, we demonstrated that CD169 RMFI in untreated patients correlated directly with the percentage of NAIVE and marginal B cells, while it correlated inversely with plasmablasts and switched B cells in all patients. These results reinforce the spatial–temporal dynamics of CD169+ macrophage activation and B-cell responses in driving antibody production. Moreover, alterations in the B-cell compartment due to SARS-CoV-2 infection may be indicative of the effort of the immune system to counterbalance lymphopenia by increasing transient B cells and plasmablasts [38,39].

Interestingly, a recent paper also demonstrated the importance of CD169 in SARS-CoV-2 infection, since it mediates antigen-presenting cells (APCs) and target cells trans-infection [40], underling the importance of the analysis of CD169 at early stage. Furthermore, here, we report the first evidence that a specific feature of the spike protein is its ability to trigger CD169 RMFI in vitro in a dose-dependent manner in parallel with enhanced IL-6 and IL-10 gene transcription in PBMCs.

Our data demonstrate that CD169 RMFI is a valuable early marker associated with SARS-CoV-2 infection and COVID-19 severity, but it is not reliable in patients who have received treatments. CD169 expression is recognized as a sensitive biomarker for other diseases, such as multiple sclerosis, in which CD169 cells promote neuro-inflammation [40,41], and systemic lupus erythematosus [42]. In these diseases, the highest levels of CD169 expression were observed in patients not receiving drugs, such as glucocorticoids or hydroxychloroquine, that decrease interferon production through the inhibition of Toll-like receptors in plasmacytoid dendritic cells [43]. Recently, lower circulating counts of CD169+ monocytes have been described in severe COVID-19 patients relative to mild cases [44]. Conversely, we observed increased CD169 expression in moderate and severe patients and in those with extensive interstitial pneumonia who did not receive any treatment.

Moreover, we also highlighted the close association between the early detection of CD169 RMFI and the respiratory outcome, suggesting that CD169 RMFI is a specific marker for patients who need respiratory support during their hospitalization.

Our findings reinforce the usefulness of CD169 RMFI at patient admission, along with T- and B-lymphocyte immunophenotyping, providing a reliable early measurement of immune status and assessment of COVID-19 disease progression to potentially drive the therapeutic approach.

## 4. Materials and Methods

### 4.1. Patients and Controls

Sixty-eight patients with positive SARS-CoV-2 RT-PCRs were enrolled in an open study by the Clinic of Infectious Diseases, Departments of System Medicine, University of Rome “Tor Vergata” and Policlinic of Tor Vergata (PTV) of Rome Foundation. Ethical approval for the collection and use of human samples was obtained from the ethical committee of “Tor Vergata” Hospital, COronaVIrus Disease: Safety and Efficacy of Experimental Treatment (COVID_SEET prot.7562/2020. 9 April 2020, experimental register 46.20). Blood samples from healthy donors (*n* = 57, HDs) were obtained from individuals attending the local blood Transfusion Unit of PTV who were referred to the Virology Unit of PTV for screening. All the HDs and COVID-19 patients provided written informed consent. SARS-CoV-2 infection was diagnosed by the Virology Unit of PTV using the Allplex 2019-nCoV Real-time PCR (SeeGene, Seoul, Korea) assay according to the manufacturer’s instructions.

Haematology and biochemical parameters of COVID-19 patients were measured on blood, serum, and plasma samples collected upon admission to the emergency department, infectious disease unit. Healthy individuals were screened for routine serology analysis. White blood cell (WBC) count was determined by using an automated hematological analyzer (Dasit-Sysmex, Milan, Italy). Serum levels of glucose (70–100) mg/dl, BUN (15–40 mg/dl), AST (5–34 U/L), ALT (0–55 U/L), LDH (125–220 U/L), C-reactive protein (CRP) (0–5 mg/L), lipase (8–78 U/L), and amylase (25–125 U/L) were measured by using an immunoturbidimetric method (Abbott Diagnostics, Milan, Italy). Plasma fibrinogen (200–400 mg/dL), D-dimer levels (0–500 ng/mL), and antithrombin III % (75–128) were measured using the Clauss method (ACL-TOP instrumentation, Werfen, Milan, Italy). Laboratory data that were collected through electronic medical records (Modulab^®^) were reported in Table 2 ((a), (b)) and in the Results in the “Demographics and Clinical Classification of COVID-19 Patients” section.

Samples were collected in the first several days of hospitalization (max. 5 days), and the clinical analysis, CT scans, immunophenotyping, CD169 determination, cytokine expression, and degree of severity refer to the same period. By contrast, the respiratory outcome applies to the entire duration of hospitalization, which may have ended with a transition to intensive care, discharge, or death, and was assessed and categorized based on the maximum need for oxygen throughout the duration of hospitalization.

### 4.2. In Vitro Stimulation with Spike Protein

Peripheral blood mononuclear cells (PBMCs) from EDTA blood samples of all the individuals enrolled in the study were isolated by density gradient centrifugation (Pancoll human) and collected immediately after separation. PBMCs were cultured at a density of 0.25 × 10^6^ in RPMI 1640 (PAN-Biotech) enriched with 2 mM L-glutamine, 100 U/mL penicillin, 0.1 mg/mL streptomycin, and 12% fetal bovine serum in the presence of 20 U/mL human recombinant interleukin-2 (IL-2) (all from Sigma, MO, USA). PBMCs were exposed to 0.1, 1, 5, and 10 nM concentrations of a SARS-CoV-2 S protein active trimer (ACROBiosystems. San Jose. CA, USA) for 24 h at 37 °C in 5% CO_2_. After incubation, the samples were recovered and analyzed by flow cytometry and real-time PCR. Each culture condition was tested in duplicate.

### 4.3. Flow Cytometry for CD169 Expression and Immunophenotyping Analysis

EDTA blood samples (10 μL) were simultaneously lysed with 500 μL of Versalyse lysing solution (Beckman Coulter) and stained with CD64-CD169/infection dried custom mixture composed of anti-CD169-phycoerythrin (PE) (clone 7-239), anti-CD64-Pacific Blue (PB), and HLA-DR (APC) (clone 22) (Beckman Coulter).

The DuraClone IM T-cell subset tube and B-cell subset tube from Beckman Coulter were used to analyze differentiation and exhaustion markers. Stained cells were then washed with Dulbecco’s phosphate-buffered saline (PBS, PAN-Biotech). The stained cells were examined using a CytoFLEX (Beckman Coulter), and the data were analyzed using the CytExpert 2.2 software (Beckman Coulter). The gating strategy is reported in Appendix A and described in previous work [12,15]. In monocytes, the expression of mCD169 MFI was very high in COVID-19 patients compared to healthy donors (*p* = 0.010), while in lymphocytes, no significant difference was detected. Interestingly, the ratio of the CD169 MFI between monocytes and lymphocytes (RMFI) was even more significant (*p* < 0.001) than the MFI in monocytes alone (Appendix A); for this reason, all the results are expressed as the RMFI. The analysis of CD169 MFI in monocytes and lymphocytes were reported in Appendix A and in Appendix A.

### 4.4. RNA Extraction from Blood Samples

Blood samples were centrifuged at 800 g for 8 min and washed with PBS two times. The obtained pellets were treated with 150 μL of Red Blood lysing buffer (GRiSP, Lda) for 5 min at room temperature two times to remove red cells. After washing with PBS, the pellets were resuspended in 400 μL of R1 buffer for RNA (GRiSP) containing 1 mM dithiothreitol (DTT, Merck, Germany), and incubated at room temperature for 5 min. Next, the samples were mixed with 70% ethanol and transferred to RNA mini spin columns according to the manufacturer’s instructions (Total RNA Extraction Kit Blood, GRiSP). Treatment with DNase I “in column” at room temperature for 15 min ensured the removal of contaminating DNA. The RNA samples were evaluated using a Nanodrop DS 11 (DeNovix. USA), showing 260/280 ratios of about 2.0 and concentrations ranging from 10 to 100 ng/μL.

### 4.5. Real-Time Quantitative RT-PCR

DNase-treated RNA (100 ng) was reverse-transcribed into cDNA using an Improm-II Reverse Transcription System (Promega, Fitchburg, Wisconsin, DC, USA) and oligo dT according to the manufacturer’s protocol; controls without the template and another without the enzyme were included in each RT reaction. A total of 2.5 ng of initial RNA in the RT reaction was used to quantitatively evaluate genes, and the gene expression of IL-6, IL-10, TNF-α, and IFN-γ was analyzed by real-time PCR (all the primer pairs used are listed in Table 11) [43,44]. The assays were performed in a Bio-Rad instrument (CFX96 Real-Time System, Bio-Rad, Hercules, CA, USA) using SYBR Green chemistry (Fast QPCR Master Mix, Smobio, Taiwan). Each sample was analyzed in triplicate, and a negative control (no template reaction) was included in each experiment to check for contamination. The expression of the housekeeping gene beta-glucuronidase (GUSB) in healthy donors was used to normalize the results. Each experiment was completed with a melting curve analysis to confirm the specificity of amplification and the absence of non-specific products and primer dimerization. Quantification was performed using the threshold cycle (Ct) comparative method: the relative expression was calculated as follows: 2 − [∆Ct (sample) − ∆Ct (calibrator) = 2 − ∆∆Ct, where ∆Ct (sample) = [Ct (target gene) − Ct (housekeeping gene)], and ∆Ct (calibrator) was the mean of ∆Ct of all the HD samples.

### 4.6. Statistical Analysis

The statistical analysis of group-wise expression levels was performed through the non-parametric Mann–Whitney test in the case of 2 independent samples or Kruskal–Wallis test and Bonferroni’s correction in the case of n-independent samples. Pearson’s correlation and Benjamini Hochberg FDR correction were used to assess the relationship between two continuous variables after testing for gaussianity through the Shapiro–Wilk test. For the hematology and biochemistry analysis, a one-sample *t*-test with respect to the range values was performed. Comparisons were considered statistically significant when *p* ≤ 0.050. Data analyses were performed using the SPSS statistical software system (version 23.0 for Windows, Los Altos, CA, USA).

## Figures and Tables

**Figure 1 pathogens-10-01639-f001:**
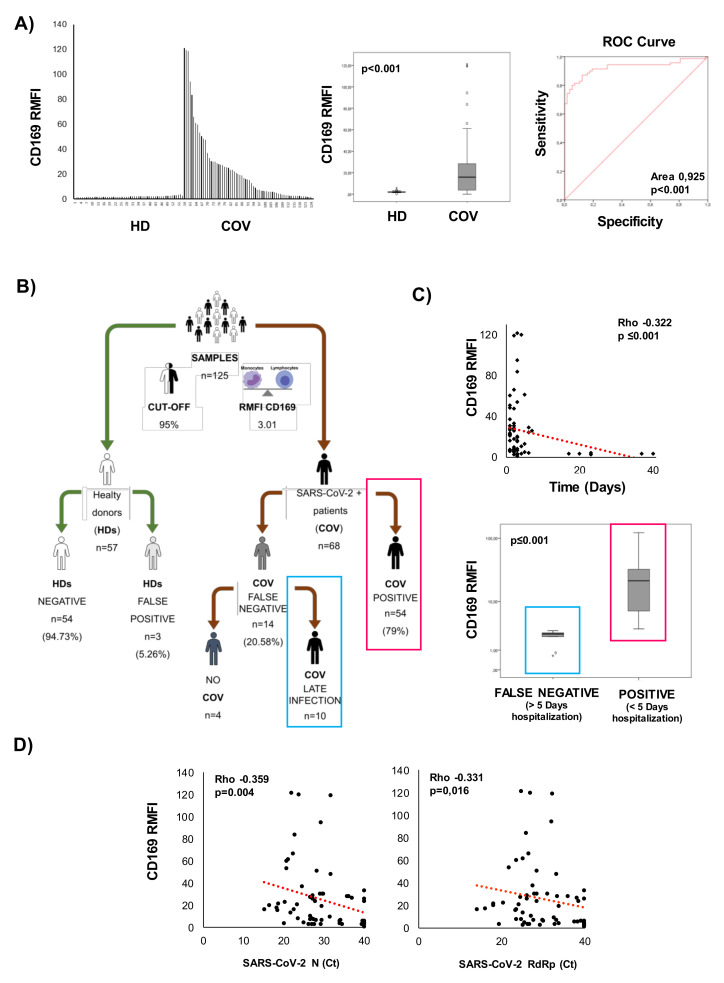
CD169 analysis by flow cytometry to discriminate COVID-19 patients. The ratio of the MFI of CD169 between monocytes and lymphocytes (RMFI) was used in the screening study as described in the Methods section and in Appendix A. (**A**) CD169 RMFI values in enrolled healthy donors (*n* = 57) and COVID-19 patients (*n* = 68); box plot of the analyzed population; ROC curve for mCD169 MFI and CD169 RMFI; the area under the ROC curve (AUC) is indicated. (**B**) Workflow of the screening carried out on CD169 expression in collaboration with Policlinic of Tor Vergata of Rome Foundation. (**C**) Relationship between days of hospitalization and CD169 expression in patients hospitalized for less than 5 days and for more than 5 days before sampling. (**D**) Scatter plot of SARS-CoV-2 N and RdRp genes detected in swab samples and represented as cycle threshold (Ct). A non-parametric Mann–Whitney test was used to compare groups, and Pearson’s correlation coefficient was calculated. Values were considered statistically significant when *p* ≤ 0.050.

**Figure 2 pathogens-10-01639-f002:**
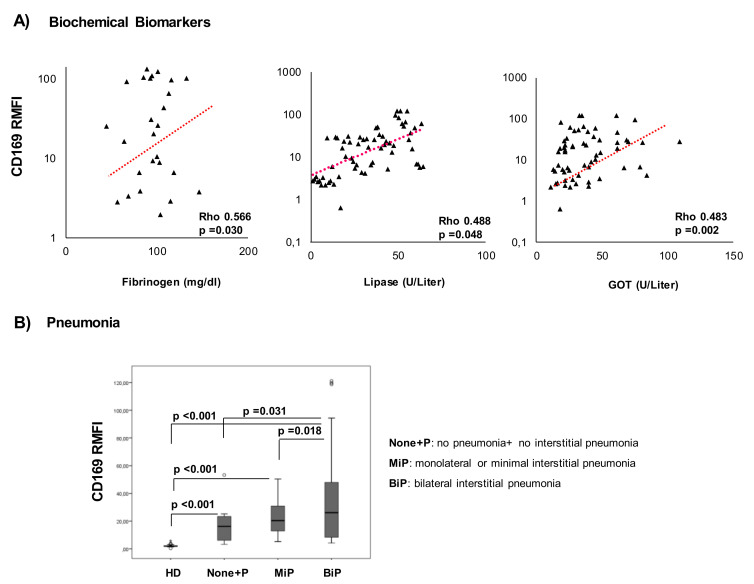
Elevated CD169 RMFI correlates with inflammatory markers and is associated with pneumonia status in COVID-19 patients. Scatter plot of (**A**) biochemical markers (X-axis) and CD169 RMFI (Y-axis) in COVID-19 patients. Among all the biochemical markers examined (Table 2 (b)), fibrinogen, lipase, and GOT correlated with CD169. (**B**) Patients were stratified into three groups based on pulmonary status and compared to HDs (*n* = 57): no pneumonia and non-interstitial pneumonia (None+P, *n* = 28), monolateral or minimal interstitial pneumonia (MiP, *n* = 6), and bilateral or severe pneumonia (BiP, *n* = 23). CD169 RMFI is represented as a box plot of all groups examined, and statistical differences are shown. Non-parametric Kruskal–Wallis tests and Bonferroni’s corrections were used to compare groups, and pairwise associations between continuous variables were tested through Pearson’s correlation coefficients. Values were considered statistically significant when *p* ≤ 0.050.

**Figure 3 pathogens-10-01639-f003:**
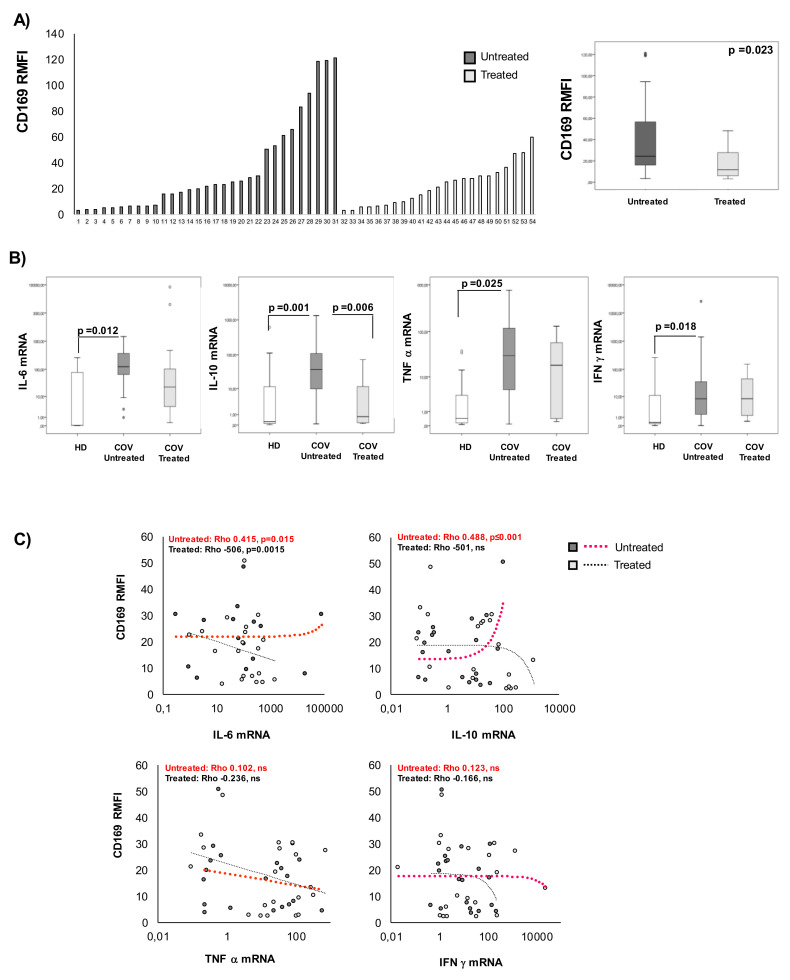
CD169 RMFI correlated with IL-6 and IL-10 in untreated COVID-19 patients. Patients were stratified into two groups based on treatment with antiviral and corticosteroids at sampling (COV-treated, *n* = 25; COV-untreated, *n* = 19) and are represented in ascending order of CD169 RFMI (left panel) (**A**) and median ± SD of CD169 RMFI in treated vs. untreated COV patients is represented as a box plot, and the statistical difference is shown (right panel). (**B**) IL-6, IL-10, IFN-γ, and TNF-α mRNA expression levels in HD (white), COV-untreated (gray), and COV-treated (light gray) are represented as box plots, and statistical differences are shown. (**C**) Scatter plots of cytokine expression (X-axis: IL-6, TNF-α, IL-10, and IFN-γ) according to real-time qRT-PCR and CD169 RMFI in COV-untreated (gray dots) and COV-treated (light-gray dots) at sampling (X-axis). Non-parametric Kruskal–Wallis tests and Bonferroni’s corrections were used to compare groups; Pearson’s correlation coefficients and Benjamini Hochberg FDR corrections were used. Values were considered statistically significant when *p* ≤ 0.050.

**Figure 4 pathogens-10-01639-f004:**
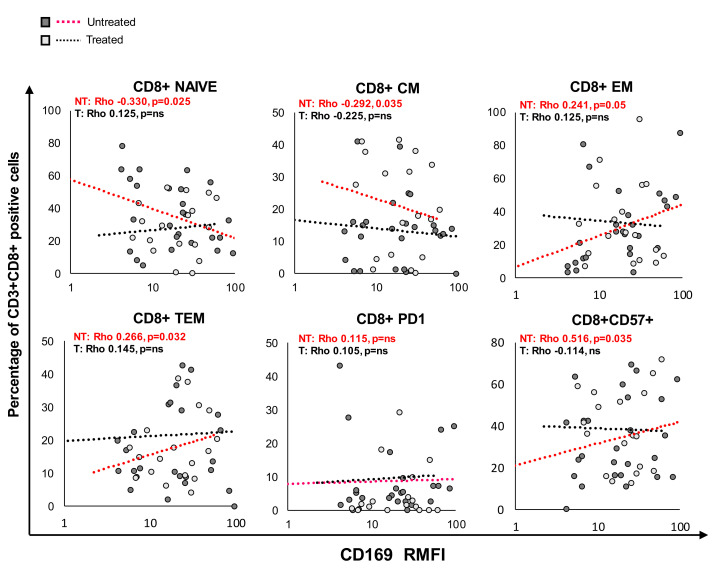
CD169 RMFI correlated with the expression of differentiation and senescence/exhaustion markers in CD8 T cells from COVID-19 patients. Patients were stratified into two groups based on treatment at sampling with antiviral + corticosteroids (treated COV, *n* = 23; untreated COV, *n* = 31). Scatter plot of CD169 RMFI (X-axis) and the expression of markers of differentiation and senescence/exhaustion in CD3+CD8+ T cells (Y-axis) in COVID-19 patients. The gating strategy to analyze markers related to differentiation, activation status, senescence, and exhaustion in T cells was provided by Beckman Coulter (Duraclone): specifically, NAIVE (CCR7+CD45RA+CD28+CD27+), central memory (CM: CCR7−CD45RA+CD28+CD27+/−), effector memory (EM: CCR7−CD45RA−CD28+/−CD27+/−), terminal effector memory (TEM: CCR7−CD45RA+CD28−CD27+/−), PD1+ exhausted, and CD57+ senescent T cells. Pearson’s correlations and Benjamini Hochberg FDR corrections were calculated. Values were considered statistically significant when *p* ≤ 0.050.

**Figure 5 pathogens-10-01639-f005:**
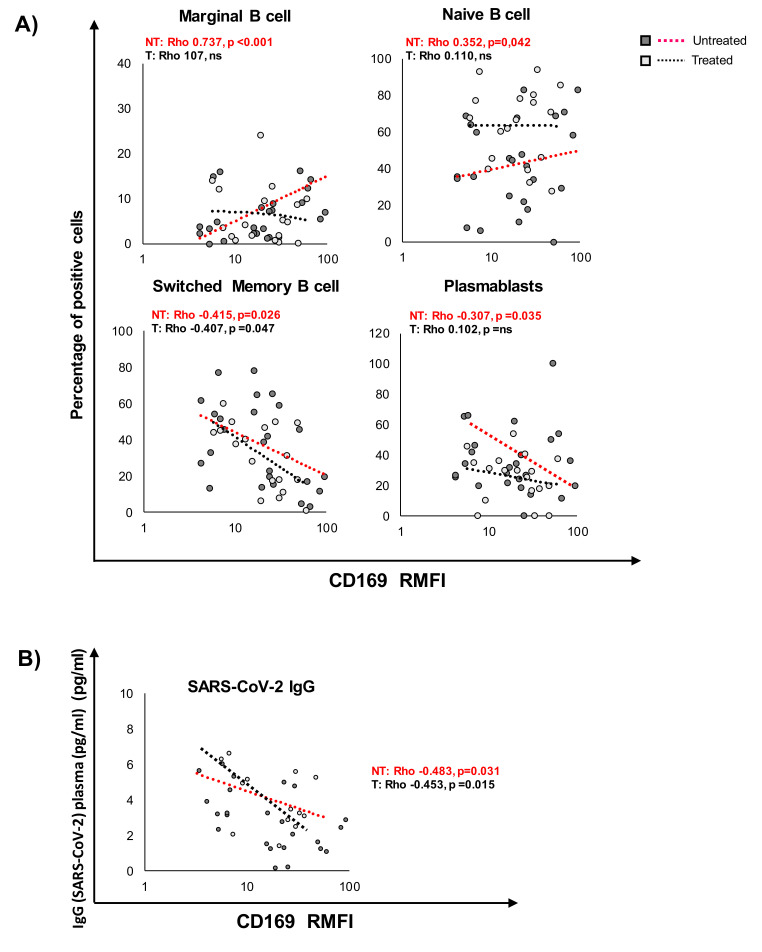
CD169 RMFI correlates with the expression of differentiation and maturation markers in B cells from COVID-19 patients and with SARS-CoV-2 IgG. Scatter plot of CD169 RMFI (X-axis) and the expression of markers of differentiation and maturation (**A**) in CD19 B cells of COVID-19 patients. The gating strategy to analyze markers related to differentiation, activation status, senescence, and exhaustion in T cells: specifically, NAIVE (CD45+CD19+CD27-IgD+), marginal (CD45+CD19+CD27+IgD+), unswitched memory (CD45+CD19+CD27+CD38-IgD+IgM+), switched memory (CD45+CD19+CD27+CD38-IgD-IgM-), and plasmablasts (CD45+CD19+CD27+CD38+IgD-IgM-). (**B**) IgG specific for SARS-CoV-2 detected in sera of COV patients at least one week after sampling. Pearson’s correlations and Benjamini Hochberg FDR corrections were calculated. Values were considered statistically significant when *p* ≤ 0.050.

**Figure 6 pathogens-10-01639-f006:**
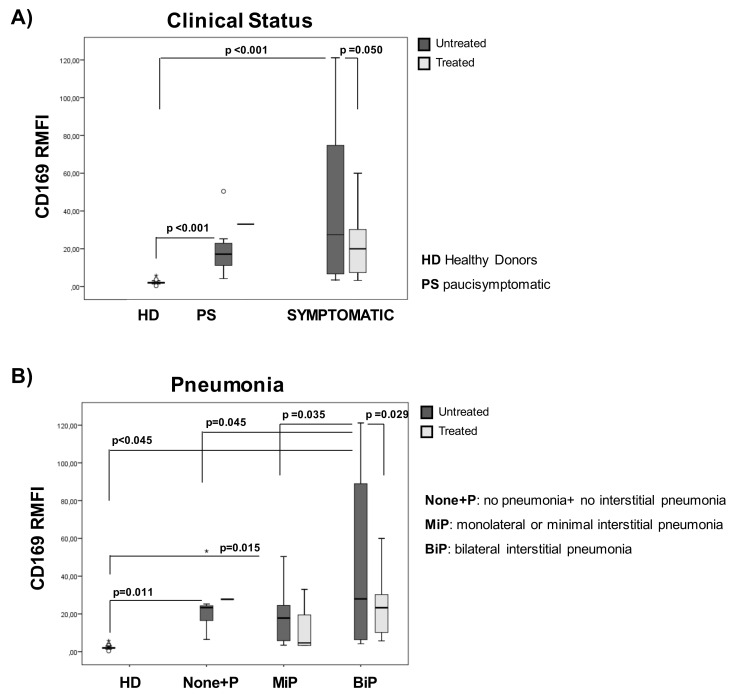
CD169 RMFI’s association with the severity of the disease and pulmonary involvement depends on the treatment at the time of sampling. COVID-19 patients (*n* = 54) were stratified into two groups according to their clinical statuses: paucisymptomatic (*n* = 19) and symptomatic (*n* = 35; mild, *n* = 13; moderate, *n* = 12; and severe, *n* = 18) (**A**). Patients were also stratified into 3 groups based on pulmonary status (**B**), as previously described (Figure 2B). CD169 RMFI is represented as box plots (gray box plots: patients positive for SARS-CoV-2 not treated at sampling (*n* = 31); light-gray box plots: patients (*n* = 23) treated with antiviral and corticosteroids at sampling; statistical differences are shown). Non-parametric Kruskal–Wallis tests and Bonferroni’s corrections were used to compare groups, and values were considered statistically significant when *p* ≤ 0.050.

**Figure 7 pathogens-10-01639-f007:**
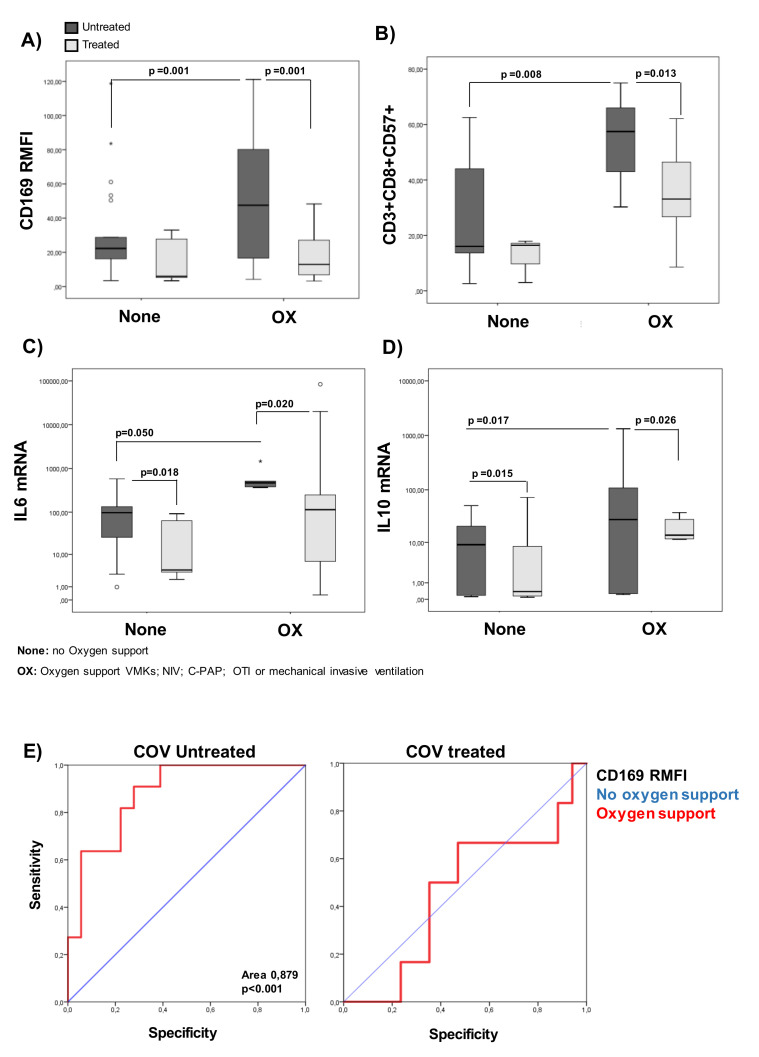
CD169 RMFI reflects respiratory outcomes of untreated COVID-19 patients. COVID-19 patients (*n* = 54) were stratified according to respiratory needs during hospitalization: no oxygen support needed (None; *n* = 27; 20 untreated and 6 treated) and oxygen support (OX; *n* = 27; 11 untreated and 17 treated). (**A**) CD169 RMFI is represented as box plots (gray box plots: patients positive for SARS-CoV-2 and not treated at sampling; light-gray box plots: patients treated with antiviral and corticosteroids at sampling). The statistical differences are shown. (**B**) The percentages of CD8 senescent cells for different treatments. (**C**) The levels of IL-6 for different treatment groups. (**D**) The levels of IL-10 expression (mRNA) for different treatment groups. (**E**) ROC curve for CD169 RMFI in untreated or treated COV with respect to oxygen support. The area under the ROC curve (AUC) is indicated. Non-parametric Kruskal–Wallis tests and Bonferroni’s corrections were used to compare groups, and values were considered statistically significant when *p* ≤ 0.050.

**Figure 8 pathogens-10-01639-f008:**
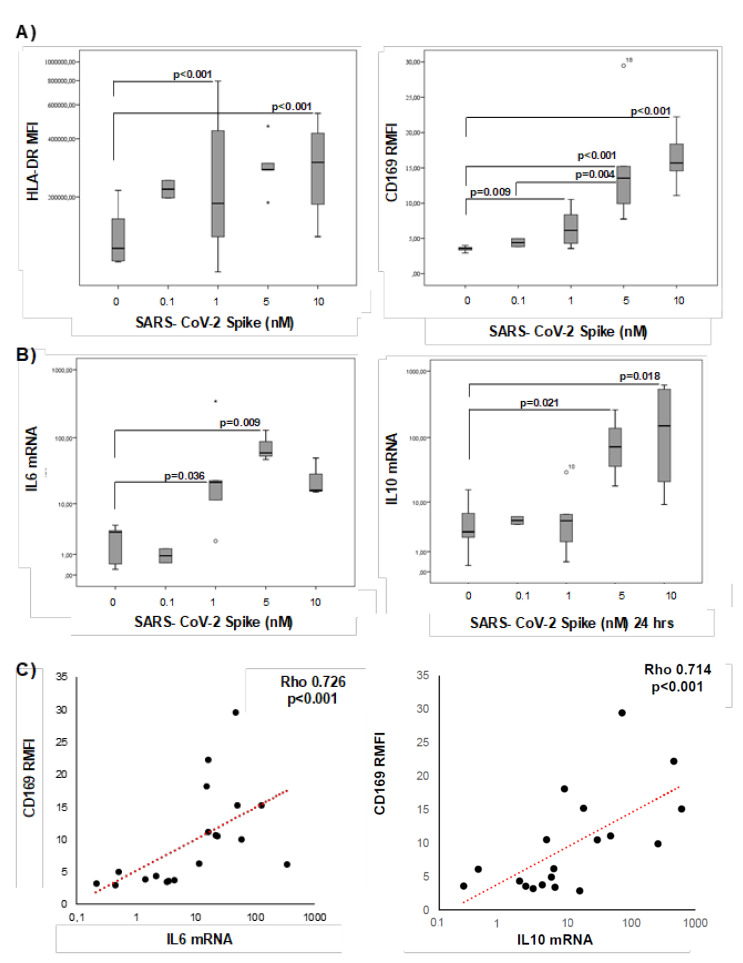
In vitro stimulation of PBMCs from healthy donors with SARS-CoV-2 induces CD169 RMFIs and correlates with the expression of IL-6 and IL-10. HLA-DR MFI and CD169 RMFI in PBMCs from healthy donors stimulated in vitro with different concentrations (range: 0-10 nM) of spike protein for 24 h are represented as box plots; statistical differences are shown (**A**). IL-6 and IL-10 mRNA expression in the same samples are represented as box plots (**B**). Scatter plot of IL-6 and IL-10 mRNA expression (X-axis) and CD169 RMFI (Y-axis) in COVID-19 patients (**C**). Non-parametric Kruskal–Wallis tests and Bonferroni’s corrections were used to compare groups. Pearson’s correlation was used to assess the relationship between the two continuous variables. Values were considered statistically significant when *p* ≤ 0.050.

**Table 1 pathogens-10-01639-t001:** Median values, interquartile ranges, and non-parametric Kruskal–Wallis and Bonferroni’s corrections of CD169 MFI in lymphocytes, monocytes, and the Ratio of CD169 MFI according to flow cytometry, in healthy donors and in COVID-19 patients.

		COV			HD		
		CD169 MFI Lymphocytes	CD169 MFI Monocytes	CD169 Ratio	CD169 MFI Lymphocytes	CD169 MFI Monocytes	CD169 Ratio
N		64	64	64	57	57	57
Median		3261.85	* 44,190.10	*** 16.65	3037.90	4353.80	2.01
IQR	25	2043.83	18,005.93	5.28	2559.60	1567.80	1.76
	50	3261.85	44,190.10	16.65	3037.90	4353.80	2.01
	75	5185.43	89,203.18	30.12	4498.90	5222.00	2.30

* and *** Significant differences according to non-parametric Kruskal–Wallis and Bonferroni’s correction vs. HD; Values were considered statistically significant when *p* ≤ 0.050; interquartile range (IQR).

**Table 2 pathogens-10-01639-t002:** Classification of COVID-19 patients at enrollment.

**(a) Demographics, clinical status at enrollment and treatment**				
		**Paucisymptomatic ***	**Symptomatic ****	**Total**
Number		19	35	54
Age (Mean ± SD)		59 ± 13	61 ± 17	60 ± 15
Sex (M/F)		12/7	29/6	41/13
Hospitalization (Days) ^+^		3.5 ± 2	6 ± 10	4.54 ± 7.54
Pneumonia	None ***	5	7	12
	P	5	11	16
	MiP	6	0	6
	BiP	6	17	23
Comorbidities	None	5	4	10
	Obesity	2	6	8
	Diabetes	3	6	9
	Cardiovascular	10	9	19
	Others ****	9	6	15
Mortality		0	6	6
No treatment		14	17	31
Treatment (Antiviral and Corticosteroid)		2	18	23
**(b) Hematology (1) and biochemistry (2) at enrollment, and respiratory outcomes (3)**				
(1) Hematology		**Paucisymptomatic**	**Symptomatic**	
Red blood cells (4.40-6.00) 10^6^ /µL		4.32 ± 0.07	4.76 ± 0.43	
Hemoglobin (13.00–18.00) g/dL		13 ± 1.36	13.68 ± 2.10	
White blood cells (4.30–10.80) 10^5^/µL		3.21 ± 1.43	7.15 ± 1.28	
Neutrophils Abs count % (40–75)		2.65 ± 1.25	5.83 ± 2.55	
		56.6 ± 17	**78.89 ± 1.83 ^++^**	
Lymphocytes Abs count % (20–45)		2.05 ± 0.23	0.07 ± 0.57	
		26.03 ± 10.00	**14.59 ± 2 ^++^**	
Monocytes Abs count % (3.4–11)		0.38 ± 0.18	0.43 ± 0.40	
		8.96 ± 5.7	6.12 ± 2.6	
Eosinophils Abs count % (0–7)		0.08 ± 0.13	0.013 ± 0.3	
		2.1 ± 3.7	1.0 ± 3.60	
Basophils Abs count % (0–1.5)		0.015 ± 0.01	0.014 ± 0.01	
		0.35 ± 0.2	0.19 ± 0.07	
(2) biochemistry		**Paucisymptomatic**	**Symptomatic**	
Fibrinogen (200–400) mg/dL		460 ± 171	**527 ± 1.3 ^++^**	
Antithrombin III % (75–128)		100 ± 3	101 ± 19	
D-dimers (0–500) ng/mL		745 ± 633	**1133 ± 91 ^++^**	
Glucose (70–100) mg/dL		101 ± 32	**119 ± 51**	
BUN (15–40) mg/dL		41 ± 28	**65 ± 40 ^++^**	
AST (5–34) U/L		27 ± 11	**44 ± 46**	
ALT (0–55) U/L		24 ± 17	**107 ± 63 ^++^**	
LDH (125–220) U/L		234 ± 71	**438 ± 21 ^++^**	
C-reactive protein (CRP) (0–5) mg/L		25 ± 22	**49 ± 49 ^++^**	
Lipase (8–78) U/L		55 ± 47	**110 ± 64 ^++^**	
Amylase (25–125) U/L		70 ± 32	110 ± 53 ^++^	
(3) Respiratory outcome ^+++^		**Paucisymptomatic**	**Symptomatic**	
None ^∞^		27	0	
NC/VMK		0	12	
NIV/C-PAP/OTI		0	15	

* Defined as paucisymptomatic (PS), including at least one COVID-19-related symptom, such as cough or fever, and not showing signs of pneumonia on physical examination. ** Defined as symptomatic: Mild (*n* = 13), Moderate (*n* = 12), and Severe (*n* = 10), as described in the text. ^+^ Days between start of hospitalization and enrollment in the study. *** None = no pneumonia, P = no interstitial pneumonia, MiP = monolateral or minimal interstitial pneumonia, BiP = bilateral interstitial pneumonia; **** other comorbidities or habits: solid organ replacement, gastrointestinal, smoking. Numbers in bold: values of analysis outside the reference ranges. One-sample *t*-tests were performed. ^++^ Mann–Whitney U test paucisymptomatic vs. symptomatic; *p* ≤ 0.01. Respiratory outcome ^+++^ (the maximum need for oxygen throughout the duration of hospitalization). ^∞^ None = no oxygen support, NC/VMK = nasal cannula/venturi mask, NIV/C-PAP = non-invasive ventilation, or OTI = invasive ventilation.

**Table 3 pathogens-10-01639-t003:** Median values, interquartile ranges, and non-parametric Kruskal–Wallis and Bonferroni’s corrections of CD169 RMFI according to flow cytometry, as well as cytokine mRNA expression in healthy donors and in untreated/treated COVID-19 patients.

**HD**		**CD169 RMFI**	**IL-6**	**IL-10**	**IFN-γ**	**TNF-α**
*n*		57	19	19	19	19
Median		1.92	0.04	0.26	0.32	0.45
IQR	25	1.58	0.03	0.10	0.18	0.16
	50	1.92	0.04	0.26	0.32	0.45
	75	2.31	81.12	12.18	12.38	3.57
**UNTREATED**		**CD169 RMFI**	**IL-6**	**IL-10**	**IFN-γ**	**TNF-α**
*n*		31	25	25	25	25
Median		**32.28**	**25.39**	**37.71**	**8.09**	**29.90**
IQR	25	6.67	4.61	9.40	1.46	4.63
	50	22.28	25.39	37.71	8.09	29.90
	75	57.21	184.28	116.95	79.86	121.76
**TREATED**		**CD169 RMFI**	**IL-6**	**IL-10**	**IFN-γ**	**TNF-α**
*n*		23	19	19	19	19
Median		**23.30 ****	**11.68**	**0.78 ****	8.19	19.05
IQR	25	9.94	5.517	0.15	1.31	0.41
	50	23.30	11.68	0.780	8.19	19.05
	75	33.99	36.92	11.57	45.84	59.82

Numbers in bold indicate significant differences according to non-parametric Kruskal–Wallis and Bonferroni’s corrections vs. HDs; ** untreated vs. treated patients. Values were considered statistically significant when *p* ≤ 0.050; interquartile range (IQR).

**Table 4 pathogens-10-01639-t004:** Median values, interquartile ranges, and non-parametric Kruskal–Wallis and Bonferroni’s corrections of the percentages of T-cell subsets in healthy donors and in untreated/treated COVID-19 patients according to flow cytometry.

**(a) T-Cell Phenotype (% of Cells)**								
**HD**		**LYMPHO**	**MONO**	**NEUTRO**	**CD3**	**CD4**	**CD8**	**CD8CD4**
*n*		57	57	57	57	57	57	57
Median		22.85	5.69	46.46	72.42	61.20	32.89	0.96
IQR	25	15.03	4.72	29.28	52.76	56.54	25.16	0.42
	50	22.85	5.69	46.46	72.42	61.20	32.89	0.96
	75	29.11	7.53	61.69	80.07	64.69	37.08	1.69
**Untreated**		**LYMPHO**	**MONO**	**NEUTRO**	**CD3**	**CD4**	**CD8**	**CD8CD4**
*n*		31	31	31	31	31	31	31
Median		**12.64**	4.34	**62.41**	**59.85**	**51.02**	29.50	0.84
IQR	25	8.86	2.86	45.70	50.39	42.80	21.95	0.32
	50	12.64	4.34	62.41	59.85	51.02	29.50	0.84
	75	16.87	6.68	73.17	76.42	65.86	37.72	2.14
**Treated**		**LYMPHO**	**MONO**	**NEUTRO**	**CD3**	**CD4**	**CD8**	**CD8CD4**
*n*		23	23	23	23	23	23	23
Median		**10.86**	6.20	**64.09**	**56.52**	**57.22**	**26.67**	1.05
IQR	25	5.89	4.77	50.54	50.93	44.78	23.43	0.56
	50	10.86	6.20	64.09	56.52	57.22	26.67	1.05
	75	18.02	8.87	78.25	68.43	69.25	33.72	2.76
**(b) Differentiation Markers in CD4+ T Cells (%)**								
**HD**		**CD4CM**	**CD4NAIVE**	**CD4EM**	**CD4TEM**	**CD4CD57**	**CD4PD1**	**CD4PD1CD57**
*n*		57	57	57	57	57	57	57
Median		50.14	38.19	50.14	0.46	3.56	1.95	0.33
IQR	25	35.85	30.17	35.85	0.24	1.66	1.15	0.13
	50	50.14	38.19	50.14	0.46	3.56	1.95	0.33
	75	62.95	48.30	62.95	0.63	7.50	4.70	0.79
**Untreated**		**CD4CM**	**CD4NAIVE**	**CD4EM**	**CD4TEM**	**CD4CD57**	**CD4PD1**	**CD4PD1CD57**
*n*		31	31	31	31	31	31	31
Median		47.58	31.30	**8.46**	0.63	3.65	**10.87**	0.96
SD		17.54	14.67	12.35	4.08	6.66	17.28	2.92
IQR	25	40.37	24.00	2.53	0.18	1.41	4.68	0.38
	50	47.58	31.30	8.46	0.63	3.65	10.87	0.96
	75	67.33	41.62	16.68	2.30	7.23	23.83	1.52
**Treated**		**CD4CM**	**CD4NAIVE**	**CD4EM**	**CD4TEM**	**CD4CD57**	**CD4PD1**	**CD4PD1CD57**
*n*		23	23	23	23	23	23	23
Median		49.89	35.94	**5.74**	**1.73**	2.66	**8.39**	0.55
IQR	25	33.85	26.88	3.28	0.24	1.54	0.17	0.22
	50	49.89	35.94	5.74	1.73	2.66	8.39	0.55
	75	62.11	46.10	17.77	4.62	10.18	21.07	1.77
**(c) Differentiation, Senescence and Exhaustion Markers in CD8+ T Cells (%)**								
**HD**		**CD8CM**	**CD8NAIVE**	**CD8EM**	**CD8TEM**	**CD8CD57**	**CD8PD1**	**CD8PD1CD57**
*n*		57	57	57	57	57	57	57
Median		27.94	46.79	10.55	5.43	1.92	3.04	0.54
IQR	25	7.30	29.34	3.17	2.67	1.58	0.72	0.29
	50	27.94	46.79	10.55	5.43	1.92	3.04	0.54
	75	45.62	56.07	33.25	14.51	2.31	12.87	6.98
**untreated**		**CD8CM**	**CD8NAIVE**	**CD8EM**	**CD8TEM**	**CD8CD57**	**CD8PD1**	**CD8PD1CD57**
*n*		31	31	31	31	31	31	31
Median		**13.32**	**30.99**	**27.73**	**17.01**	**30.26**	5.30	2.48
IQR	25	4.84	15.26	13.38	9.04	15.04	2.59	1.35
	50	13.32	30.99	27.73	17.01	30.26	5.30	2.48
	75	15.85	52.94	47.60	31.18	51.47	10.26	6.13
**treated**		**CD8CM**	**CD8NAIVE**	**CD8EM**	**CD8TEM**	**CD8CD57**	**CD8PD1**	**CD8PD1CD57**
*n*		23	23	23	23	23	23	23
Median		**19.00 ****	**28.55**	**26.47**	**15.84**	**32.83 ****	1.53	1.98
IQR	25	5.02	16.58	12.49	9.16	24.23	0.00	0.00
	50	19.00	28.55	26.47	15.84	32.83	1.53	1.98
	75	35.00	43.95	55.51	24.50	51.39	11.03	15.47

Numbers in **bold** indicate significant differences according to non-parametric Kruskal–Wallis and Bonferroni’s corrections vs. HDs; ** untreated vs. treated. Values were considered statistically significant when *p* ≤ 0.050; interquartile range (IQR).

**Table 5 pathogens-10-01639-t005:** Pearson’s correlation between CD169 RMFI and T-cell differentiation and senescence/exhaustion markers in untreated and treated COVID-19 patients.

	Untreated		Treated	
	CD4	CD8	CD4	CD8
NAÏVE	ns	−0.330. *p* = 0.005	ns	ns
CM	0.241. *p* = 0.044	−0.292. *p* = 0.014	ns	ns
EM	ns	0.241. *p* = 0.044	ns	ns
TEM	ns	0.266. *p* = 0.026	ns	ns
Senescent (CD57+)	ns	0.516. *p* = 0.014	ns	ns
PD1+	ns	ns	ns	ns
Exhausted (CD57 +PD1+)	ns	ns	ns	ns
	CD169 RMFI		CD169 RMFI	

Negative coefficient, *p* ≤ 0.050 = green; Positive coefficient, *p* ≤ 0.050 = red; No difference, *p* ≤ 0.050 = gray.

**Table 6 pathogens-10-01639-t006:** Median values, interquartile ranges, and non-parametric Kruskal–Wallis and Bonferroni’s corrections of the percentages of B-cell subsets in healthy donors and in untreated/treated COVID-19 patients.

**Healthy Donors**		**Marginal**	**Naïve B**	**Unswitched**	**Switched** **Memory**	**Plasmablast**
*n*	57	57	57	57	57	57
Median		14.09	53.36	37.97	33.33	14.66
IQR	25	1.74	19.32	15.83	21.07	8.88
	50	14.09	39.58	30.97	33.33	14.66
	75	19.33	60.63	43.75	51.17	30.28
**Untreated COV**		**Marginal**	**Naïve B**	**Unswitched**	**Switched** **Memory**	**Plasmablast**
*n*		31	31	31	31	31
Median		**6.24**	**43.16**	*** 33.78**	**23.33**	17.86
IQR	25	2.39	26.47	16.21	22.16	10.27
	50	6.24	43.16	31.78	33.33	17.86
	75	13.28	63.08	55.44	45.13	25.84
**Treated COV**		**Marginal**	**Naïve B**	**Unswitched**	**Switched** **Memory**	**Plasmablast**
*n*		23	23	23	23	23
Median		**4.58**	**47.20**	**31.49**	**26.22**	20.67
IQR	25	1.44	44.31	16.00	14.90	8.66
	50	4.58	67.20	34.92	29.22	20.67
	75	10.49	78.80	47.64	36.38	32.23

Numbers in bold: significant differences according to non-parametric Kruskal–Wallis and Bonferroni’s corrections vs. HDs; * Untreated vs. treated. Values were considered statistically significant when *p* ≤ 0.050; interquartile range (IQR).

**Table 7 pathogens-10-01639-t007:** Pearson’s correlation between CD169 RMFI and B-cell differentiation markers in untreated and treated COVID-19 patients.

	Untreated	Treated
	CD19	CD19
Marginal	0.737. *p* < 0.001	ns
NAIVE	0.352. *p* = 0.032	ns
Unswitched	ns	ns
Switched memory	−0.415. *p* = 0.016	−0.407. *p* = 0.041
Plasmablasts	−0.307. *p* = 0.025	ns
	CD169 RMFI	

Negative coefficient, *p* ≤ 0.050 = green; Positive coefficient, *p* ≤0.050 = red; No difference, *p* ≤ 0.050 = gray.

**Table 8 pathogens-10-01639-t008:** Median values, interquartile ranges, and non-parametric Kruskal–Wallis and Bonferroni’s corrections of CD169 RMFI in healthy donors categorized according to clinical status (**a**) and pulmonary involvement (**b**).

**(a) Clinical Status**								
			**UNTREATED**			**TREATED**		
		**HD**	**PS**	**SYMPTOMATIC**		**PS**	**SYMPTOMATIC**	
*n*		57	11	20		1	22	
Median		2.00	**17.00**	**27.50**		33.00	* 20.00	
IQR	25	2.00	6.00	6.25		33.00	7.00	
	50	2.00	17.00	27.50		33.00	20.00	
	75	2.00	23.00	79.50		33.00	30.00	
**(b) Pulmonary Involvement**								
			**UNTREATED**			**TREATED**		
		**HD**	**PS**	**MiP**	**BiP**	**PS**	**MiP**	**BiP**
*n*		57	7	8	16	1	4	18
Median		2.01	**23.35**	**17.81**	** ^∞,^ ** *** 27.97**	27.74	4.67	23.30
IQR	25	1.76	15.88	5.55	6.13	27.74	3.26	9.94
	50	2.01	23.35	17.81	27.97	27.74	4.67	23.30
	75	2.30	25.27	26.66	91.66	27.74	26.24	31.93

Numbers in bold indicate significant differences according to non-parametric Kruskal–Wallis and Bonferroni’s corrections vs. HDs; * in untreated BiP vs. MiP; **^∞^** in untreated BiP vs. NONE+P; Values were considered statistically significant when *p* ≤ 0.050; interquartile range (IQR).

**Table 9 pathogens-10-01639-t009:** Median values, interquartile ranges, and non-parametric Kruskal–Wallis and Bonferroni’s corrections of the percentages of CD8CD57 cells, CD169 RMFI, IL-6 and IL-10 in untreated/treated COVID-19 patients according to oxygen need.

		None				OX			
**UNTREATED**		**CD8CD57**	**CD169**	**IL6**	**IL10**	**CD8CD57**	**CD169**	**IL6**	**IL10**
*n*		20	20	20	20	11	11	11	11
Median		16.04	21.32	94.40	0.33	**52.58**	**26.07**	**125.95**	**11.51**
IQR	25	13.33	8.81	23.51	0.11	35.65	5.88	8.92	1.18
	50	16.04	21.32	94.44	0.33	52.58	26.07	125.95	11.51
	75	36.56	44.99	152.76	7.23	63.71	94.35	515.66	27.91
**TREATED**		**CD8CD57**	**CD169**	**IL6**	**IL10**	**CD8CD57**	**CD169**	**IL6**	**IL10**
*n*		6	6	6	6	17	17	17	17
Median		23.53	24.39	** 3.82	20.77	** 32.83	*** 18.90	**87.927**	* 7.52
IQR	25	3.00	5.30	2.62	00.37	25.87	8.2872	6.60	0.26
	50	23.53	24.39	3.82	200.77	32.83	18.90	**87.92**	7.52
	75		29.27	211.91	4.25	52.39	33.61	469.59	10.84

Numbers in bold define significant differences according to non-parametric Kruskal–Wallis and Bonferroni’s corrections Oxygen needs (OX) vs. None; *p* < 0.01 **, *p* < 0.001 *** untreated vs. treated group. Values were considered statistically significant when *p* ≤ 0.050; interquartile range (IQR).

**Table 10 pathogens-10-01639-t010:** ROC curve for CD169 RMFI, IL-6, and IL-10 mRNA, and CD8+CD57+ in untreated or treated COV with respect to oxygen support. The area under the ROC curve (AUC) is indicated, as is the sensitivity and specificity of the marker analyzed. *** *p* < 0.001 in oxygen support vs. none.

OX vs. None	Untreated			Treated		
	AUC	Sensitivity (%)	Specificity (%)	AUC	Sensitivity (%)	Specificity (%)
CD169 RMFI	0.879 ***	89	80	0.656	45	33
IL-6	0.659	52	68	0.574	48	51
IL-10	0.623	53	69	0.632	51	54
CD8+CD57+	0.540	65	45	0.375	31	38

**Table 11 pathogens-10-01639-t011:** Primer pair sequences used in the real-time PCR analysis.

Gene		Forward Primer (5′→3′)	Reverse Primer (5′→3′)
GUSB	NM_000181	CAGTTCCCTCCAGCTTCAATG	ACCCAGCCGACAAAATGC
IL-6	NM_000600.3	TGCAATAACCACCCCTGACC	ATTTGCCGAAGAGCCCTCAG
IL-10	NM_000572.2	ACATCAAGGCGCATGTGAAC	CACGGCCTTGCTCTTGTTTT
TNF-α	NM_000594.3	CCCGAGTGACAAGCCTGTAG	TGAGGTACAGGCCCTCTGAT
IFN-γ	NM_000619.2	TCAGCTCTGCATCGTTTTGG	GTTCCATTATCCGCTACATCTGAA

## Data Availability

Data is contained within the article and supplementary material.

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
