# Peer review of "High CD169 Monocyte/Lymphocyte Ratio Reflects Immunophenotype Disruption and Oxygen Need in COVID-19 Patients"

_pathogens, 2021, doi:10.3390/pathogens10121639_

Round 1

Reviewer 1 Report

The authors have responded to each of the concerns raised about their initial submission, revised their statistical analysis, and have added additional data on S protein that augments their findings. But need to minor English language editing.

Author Response

We thank you for your comments; certainly, all the comments and suggestions received previously, have enriched the work. Regarding the language issue, the present manuscript underwent English language revision and was certified by MDPI language service. The certification has been attached as supplemental material not publishable in the resubmission form. However, we have re-read the text carefully and corrected it where required.

Reviewer 2 Report

In this manuscript, authors have shown that the ratio of mean fluorescence intensity (RMFI) of CD169 on monocytes/lymphocytes can be used as a biomarker to evaluate the immune modulation and oxygen requirements in COVID-19 patients, especially in untreated patients. The authors have made most of the relevant comparisons and the findings have important implications for COVID-19 patients. However, there are some concerns and suggestions:

  1. Authors have shown only the MFI of CD169 but not the number of CD169+ Since, earlier it has been shown that severe COVID-19 patients have lower circulating counts of CD169+monocytes compared to mild cases, it would be important to show if only CD169 MFI can serve as a biomarker for disease severity and not the counts of CD169monocytes or if both can be used as biomarkers and if there are any differences between the two.
  2. It will be more informative and useful to show the relation of CD169 RMFI with the severity of disease. Therefore, in the symptomatic patient group, authors should also show relation of CD69RMFI with mild, moderate and severe disease separately.
  3. There are certain inconsistencies between the data shown in the tables/figures and mentioned in the text. Few are listed below:
  • Table 3 and lines 205-206: It is mentioned that “……while in treated patients, IL-6 and IL-10 had significantly lower values than those of untreated patients.” However, IL-6 has a value of 116.85 in treated patients in table 3, which is much higher than 25.39 given for the untreated patients.
  • Lines 234-235: It is mentioned that “…..and a significant increase in CD4+cells expressing markers of senescence and exhaustion (CD57+/PD1+) was detected only in treated patients.” However, neither the double positive (CD57+/PD1+) nor single positive cells are bold in the table 4 (that shows statistical significance).
  • Lines 236-240: Authors mention that there is a decrease in CD8CM only in untreated patients compared to healthy donors (HD) - however, this is not shown as significant in the table; similar decrease is seen in treated patients also (a median value of 19 vs 27.94, not only in untreated patients as mentioned by the authors). Furthermore, there is a significant decrease in CD8naive cells in both untreated and treated patients as shown in the table 4 but not mentioned in the text.
  • Figure 4 legend: CD4 data is not shown in the figure but mentioned in the legend as “A”.
  • Table 9: Statistical significances are not shown at all.
  1. It would be important to show the changes in the cytokines, especially IL-10 and IL-6 at protein level.

In addition, minor concerns and suggestions are:

  1. Line 211: This should be Fig. 3C (not 3B).
  2. Table 4a: For untreated patients- What are CD3p, CD4p, CD8per; Table 4b: For treated patients- what is CD4CMp; and Table 4c: For treated patients- what is CD8CD57P?
  3. Figure 5: Refer as 5A and 5B (on lines 277-285).
  4. Table 10: What do three stars represent? AUC-untreated: Replace commas with decimals; Why sensitivity and specificity are not given for IL-10 and CD8+CD57+for treated patients?
  5. Line 362: it should be Fig. 8A. Also refer 8B and 8C at relevant places.
  6. Fig. S1D- what do histograms at LHS and RHS represent?
  7. Fig. S2- The labels on the Y-axis are cut (not seen properly). What do the numbers on the bars represent?
  8. Methods for biochemistry are not provided.

Author Response

In this manuscript, the authors have shown that the ratio of mean fluorescence intensity (RMFI) of CD169 on monocytes/lymphocytes can be used as a biomarker to evaluate the immune modulation and oxygen requirements in COVID-19 patients, especially in untreated patients. The authors have made most of the relevant comparisons and the findings have important implications for COVID-19 patients. However, there are some concerns and suggestions:

  1. Authors have shown only the MFI of CD169 but not the number of CD169+Since, earlier it has been shown that severe COVID-19 patients have lower circulating counts of CD169+monocytes compared to mild cases, it would be important to show if only CD169 MFI can serve as a biomarker for disease severity and not the counts of CD169monocytes or if both can be used as biomarkers and if there are any differences between the two.

Based on the bibliography which contains a majority of articles describing CD169 RMFI in this context, rather than CD169 cell number, we have designed the study to measure CD169 fluorescence. Measuring such an absolute cell count accurately would have required the addition of calibration beads (such as Flowcount beads), and therefore we cannot extract reliable numbers out of our current data. However, since we have standardized the volume of blood (80ul) stained for this cohort, we have tried to analyze a relative count of circulating CD169+ monocytes/macrophages, and we confirmed that a higher number was found in COVID-19 with respect to HDs, but no significant correlation was found among CD169+ number and the complexity of the immune system dysfunction, inflammatory markers and clinical aspects found with the CD169RMFI. Furthermore, we consider it easier in the clinical routine to measure this fluorescence ratio (independent of cytometer settings, and applicable to fingerprick samples), rather than cell numbers (requiring a precise pipetting of venous blood and calibration beads).”

  1. It will be more informative and useful to show the relation of CD169 RMFI with the severity of the disease. Therefore, in the symptomatic patient group, authors should also show the relation of CD69RMFI with mild, moderate, and severe disease separately.

The methodological approach for patients’ recruitment and clinical evaluation was based on a previous study on COVID-19 patients published by us in recognized journals. Moreover, in the first version of the manuscript submitted to Pathogens, the study reported more detailed categories (pauci symptomatic, moderate and severe), but, due to the number of patients for category, the statistical analysis were less robust, and the reviewers of the first round of revision strongly suggested to divided the patients’ cohort into asymptomatics and paucisymptomatics (those who had only mild symptoms - cough or fever) and those who had more specific symptoms including Mild+Mod+Severe (with pneumonia and need for respiratory support at different levels).

Here we report the previous version of the Figure:

  1. There are certain inconsistencies between the data shown in the tables/figures and mentioned in the text. Few are listed below:
  • Table 3 and lines 205-206: It is mentioned that “……while in treated patients, IL-6 and IL-10 had significantly lower values than those of untreated patients.” However, IL-6 has a value of 116.85 in treated patients in table 3, which is much higher than 25.39 given for the untreated patients.

Thanks for the clarification. Reviewing the table, we have realized that the data had been incorrectly reported, in fact, the decimal point was shifted for all values of IL6 and IL10, the actual value is therefore 11.685 and not 116.85 in the untreated. The table has been modified with the new data.

  • Numbers Lines 234-235: It is mentioned that “…..and a significant increase in CD4+cells expressing markers of senescence and exhaustion (CD57+/PD1+) was detected only in treated patients.” However, neither the double-positive (CD57+/PD1+) nor single positive cells are bold in table 4 (that shows statistical significance).

We have carefully read the text and the table and indeed, significant differences are evident for PD1 expression in CD4 cells and not for CD57. So the text has been changed to "a significant increase in CD4+cells expressing markers of exhaustion (PD1+)" (lines 234-235).

  • Lines 236-240: Authors mention that there is a decrease in CD8CM only in untreated patients compared to healthy donors (HD) - however, this is not shown as significant in the table; a similar decrease is seen in treated patients also (a median value of 19 vs 27.94, not only in untreated patients as mentioned by the authors).

Thank you for your comment, the significant value was added in the table which, actually, only pertains to the untreated group. While the observed decrease of CD8CM in the treated was not statistically significant. This result has been added in the text (lines 277-278).

  • Furthermore, there is a significant decrease in CD8naive cells in both untreated and treated patients as shown in table 4 but not mentioned in the text.

Thank you for the clarification, the result of the CD8naive has been added in the text lines 280-281

  • Figure 4 legend: CD4 data is not shown in the figure but mentioned in the legend as “A”.

In fact, it is a refusal of a previous version of the work and has been modified as suggested.

  • Table 9: Statistical significances are not shown at all.

As suggested, significant values have been highlighted using bold or asterisks as in the other figures and the table legend has been revised.

  1. It would be important to show the changes in the cytokines, especially IL-10 and IL-6 at protein level.

Thank you for the suggestion. We are aware that the determination of these cytokines at the protein level could improve the meaning of certain scientific evidence. Although the determination of IL-6 protein was routinely on these patients at the hospital, the data are not available for all the patients. Unfortunately, we have no residual material for the determination of these cytokines.

In addition, minor concerns and suggestions are:

  1. Line 211: This should be Fig. 3C (not 3B).

We have modified as suggested

  1. Table 4a: For untreated patients- What are CD3p, CD4p, CD8per; Table 4b: For treated patients- what is CD4CMp; and Table 4c: For treated patients- what is CD8CD57P?

The letter p is a mistake of a previous version, it was all standardized and put in the title that the results are the % of leucocyte subpopulations observed or the % of positive cells observed.

  1. Figure 5: Refer as 5A and 5B (on lines 277-285).

We have modified as suggested

  1. Table 10: What do three stars represent? AUC-untreated: Replace commas with decimals; Why sensitivity and specificity are not given for IL-10 and CD8+CD57+for treated patients?

Thanks for the clarification, we have corrected the punctuation errors and added the missing numbers.

  1. Line 362: it should be Fig. 8A. Also refer 8B and 8C at relevant places.

We have modified as suggested

  1. S1D- what do histograms at LHS and RHS represent?

On the left of the figure we have the histogram of CD169MFI in monocytes and lymphocytes of a healthy donor and on the right of a COV patient. Below, the histograms referred to the MFIratio (RMFI) between monocytes and lymphocytes. We have modified the figure to better clarify this analysis

  1. S2- The labels on the Y-axis are cut (not seen properly). What do the numbers on the bars represent?

We have modified the image to better read the scale representing the MFI of CD169 in monocytes, as requested

  1. Methods for biochemistry are not provided.

Hematology and biochemical parameters of COVID-19 patients were measured on blood, serum, and plasma samples collected upon admission to the emergency department, infectious disease unit. healthy individuals were screened for routine serology analysis. Methods have been added in section 4.1 patients and controls, lines 479-489

Round 2

Reviewer 2 Report

Authors have satisfactorily responded to all the suggestions/comments. There are few minor concerns that are not addressed:

  1. Lines 237-238: Change language from: "...while in the treated group the decrease results no significant." to: while in the treated group, the observed decrease was not statistically significant.
  2. Table 9: authors have defined three stars (***) but not two stars (**).
  3. On line 367, refer Fig. 8C.
  4. For Fig. S2, also describe what are the numbers on the bars represent.

Author Response

We thank the reviewer for his comprehensive comments and suggestions. We have revised all the points as suggested, and all the amendments have been highlighted in the manuscript file.